# Explainable Machine Learning for LoRaWAN Link Budget Analysis and Modeling

**DOI:** 10.3390/s24030860

**Published:** 2024-01-29

**Authors:** Salaheddin Hosseinzadeh, Moses Ashawa, Nsikak Owoh, Hadi Larijani, Krystyna Curtis

**Affiliations:** Department of Cybersecurity and Networks, Glasgow Caledonian University, Glasgow G4 0BA, UK; moses.ashawa@gcu.ac.uk (M.A.); krystyna.curtis@gcu.ac.uk (K.C.)

**Keywords:** LoRaWAN, Internet of Things, artificial intelligence, machine learning, regression analysis, propagation modeling, link budget analysis

## Abstract

This article explores the convergence of artificial intelligence and its challenges for precise planning of LoRa networks. It examines machine learning algorithms in conjunction with empirically collected data to develop an effective propagation model for LoRaWAN. We propose decoupling feature extraction and regression analysis, which facilitates training data requirements. In our comparative analysis, decision-tree-based gradient boosting achieved the lowest root-mean-squared error of 5.53 dBm. Another advantage of this model is its interpretability, which is exploited to qualitatively observe the governing propagation mechanisms. This approach provides a unique opportunity to practically understand the dependence of signal strength on other variables. The analysis revealed a 1.5 dBm sensitivity improvement as the LoR’s spreading factor changed from 7 to 12. The impact of clutter was revealed to be highly non-linear, with high attenuations as clutter increased until a certain point, after which it became ineffective. The outcome of this work leads to a more accurate estimation and a better understanding of the LoRa’s propagation. Consequently, mitigating the challenges associated with large-scale and dense LoRaWAN deployments, enabling improved link budget analysis, interference management, quality of service, scalability, and energy efficiency of Internet of Things networks.

## 1. Introduction

The Internet of Things (IoT) and Artificial Intelligence (AI) are at the forefront of the fast-growing technology. The growth in IoT devices has been extraordinary, with a significant surge in recent years, where projections suggest over 75 billion interconnected devices by 2025 [1]. The deployment of Artificial Intelligence (AI) has led to its more profound integration into daily life, intertwining closely with technologies like the Internet of Things (IoT) [2]. The widespread adoption and deployment of IoT devices pose many challenges, and it necessitates thorough and meticulous planning to ensure the smooth and uninterrupted operation of these extensive networks [3]. It is established in [4] that a dense deployment of IoT devices may have high levels of interference and result in packet loss. This planning overlooks several aspects of the IoT networks, such as seamless functionality by minimizing dead zones, interference mitigation and quality of service, cost optimization, scalability, and energy efficiency. Zhang et al. [5] state that enhancing the transmission performance is a key issue that should be resolved by effectively optimizing the resource allocation in the LoRa network [6], and that can be achieved by adjusting the transmission parameters.

Rida et al. [7] characterize long-range communication, low data rate transmission, low energy consumption, and cost-effectiveness as the main requirements of IoT applications, all of which are present in Low-Power Wide Area Networks (LPWAN) technologies. LoRaWAN, Sigfox, NB-IoT, LTE-M, and EC-GSM are some of the well-established technologies in this category, with LoRaWAN being the most cost-efficient, especially where device density is low. A comprehensive comparison of LPWAN technologies is provided in [8].

One of the main advantages of LoRa, which has contributed to its popularity and extensive utilization, is its open-source technology, which enables autonomous and cost-effective setup [9]. Additionally, the spreading factor (SF) of the proprietary chirp spread spectrum (CSS) modulation further improves the overall sensitivity of the device and increases interference immunity [10]. The impact of SF on the data rate and time-on-air can be precisely calculated [11], whereas its impact on propagation range and the link budget requires empirical experimentation. For instance, Hosseinzadeh et al. [10] observed an equivalent range extension of about 500 m when switching from SF 7 to SF 12 in an urban area. Additionally, adjusting the transmission power and SF can directly impact the per-bit energy consumption of the LoRa network [12].

To fully understand the behavior of CSS and SF in various environments, researchers have conducted several empirical measurement campaigns in indoor environments [13,14,15,16], outdoor urban and rural areas [7,17], and even on the sea with line of sight (LoS) [18]. These empirical measurements are then compared and analyzed against propagation models to establish a model for LoRa and estimate the coverage using the received signal strength (RSS). Therefore, it is crucial to use accurate models to estimate and plan the network.

Traditional empirical models, often used in this context, were initially developed for technologies and frequencies different from those used in LoRa. For instance, models like the Log-distance and Okumura-Hata are employed in simulation tools such as LoRaSim, SimLoRaSF, LoRWAN-Sim, and the NS3-Module for their simplicity [11]. These models take a simplistic approach to characterize the propagation in the environment [19]. However, in recent years, many have taken advantage of AI to develop innovative complex models to enhance their performance. Wong et al. [20] highlight that even the AI-based models show limited ability to generalize to new hardware effects and channel conditions across different transmitter and receiver pairs. 

The main contributions of this article are:(a)Critical analysis of the latest AI advancements in the area of propagation modeling.(b)Comparing the efficiency of AI algorithms for propagation modeling using LoRaWAN practical measurements from multiple transmitters.(c)Develop an explainable and efficient AI-based model for LoRaWAN propagation estimation.(d)Using the explainability of AI to provide further insight into deterministic feature engineering and propagation mechanisms of LoRa.

The rest of this article is organized as follows: Section 2 provides a critical overview of the AI-driven propagation models by segregating and comparing machine learning and deep learning techniques. Section 3 explains the implementation details of the data collection, model selection, data analysis, and results. Section 4 concludes and summarizes the findings of the research.

## 2. AI for Propagation Modeling

Using AI for propagation modeling offers several advantages when compared to empirical models [21], such as:Adaptability: can adapt to different scenarios and technologies such as LPWAN and particularly LoRaWAN, making them suitable for various applications, from urban to rural areas and indoor spaces;Accuracy: can outperform empirical models in their accuracy as they learn from real-world data while using environmental information to their advantage;Cost and time efficiency: can reduce the time required to generate a propagation estimation.

Based on the method of feature extraction that is used, this section divides the AI into machine learning and deep learning algorithms and addresses each individually.

### 2.1. Machine Learning for Propagation Modeling

Modeling the path loss to estimate the RSS is essentially a regression problem; although AI is also used for scenario identification, such as LoS and non-LoS (NLoS), this is not within the scope of this research, and it is further studied in [21]. Therefore, researchers have experimented with several regression-based models, the most popular being the multilayer perceptrons (MLP), which is a type of artificial neural network (ANN) [22]. For instance, Hornik et al. [23] proves that a multilayer perceptron with three hidden layers can approximate an arbitrary continuous function. 

Authors in [24,25] trained an MLP using empirical measurements and environmental features such as the position and gain of the transmitter (Tx) and receiver (Rx) antennas, maximum transmission power, angle of arrival (AoA), and departure (AoD) as the input features of their models. These features are extracted using deterministic approaches applied before the regression training. Authors of [10] deployed image processing to extract environmental features, such as base station altitude, AoA and AoD, clutter, number of buildings on the LoS path, NLoS to LoS ratio, street width, and street direction from GoogleMaps to train an MLP.

However, Hosseinzadeh et al. [26] tried to address two of the shortfalls of using MLPs, namely, (a) collecting a large number of data samples that are required to train the MLP and (b) overfitting the MLP and losing generalization of predictions. This was performed using a hybrid model for LoRaWAN that combined COST231 in conjunction with an MLP, leading to a smaller number of neurons in the hidden layers. In this design, the MLP only had to complement the COST231 model rather than learning the propagation from scratch. This also reduced the number of required data samples and prevented overfitting. Using this approach, authors were able to create a predictive path-loss model that accepted LoRa’s SF as one of the input features, and therefore, the SF could have been modeled to have a direct impact on the estimation. 

To simplify the data collection campaigns, authors in [27] suggested an adaptive neuro-fuzzy model that allows embedding an expert’s knowledge into the system with the ability to learn and adjust its parameters. This achieved similar results to MLPs by utilizing only 30% of the data; this 70% reduction directly translates into a less exhaustive data collection campaign and saves significant time and cost. Authors in [28] also recommended the use of relevance vector machine (RVM) compared to other AI techniques when the volume of training samples is limited and noted a higher estimation accuracy for k-nearest neighbor (KNN) and support vector machine (SVM) algorithms compared to RVM.

### 2.2. Deep Learning for Propagation Modeling

To improve feature extraction in the AI-based propagation models, researchers have incorporated topographical characteristics of the outdoor environments (terrain features) into the propagation models. Authors in [29] recognize that empirical models only coarsely characterize the environment, and they can be inaccurate. Ray tracing solutions [30], although more accurate, have a higher computational complexity and are inflexible when utilizing global information system (GIS) maps. Hence, authors in [29,31,32,33,34] proposed deep learning (DL) through the use of the convolutional neural network (CNN) that can take GIS images of the propagation environment as input and automate the extraction of the topographical features to improve path loss estimation. The images can be collected from GIS, QGIS, Open-StreetMap, GoogleMaps, or similar platforms and could include terrain elevation, land cover, and clutter, or they could be custom-made images [32]. During the training process, DL-based models extract the features from these images in a “black-box” approach. Since there are no deterministic rules as to what constitutes helpful features, there is no transparency and traceability in their decision-making.

The regression part of the problem has been addressed by using U-Net architecture [29,31], whereas others have used MLPs. The feature maps from the last layer of the CNN are used as the inputs of the MLP to predict propagation or path loss. Each pixel within the feature map would become the input of the MLP, and the measured RSS serves as the ground truth output. Following several training iterations, updating the feature maps and MLP parameters (weights and biases), the model would be able to predict the propagation, given sets of 2D images as input. This approach essentially trains two sub-models, one of which is the CNN, trained to extract relevant features from 2D images, and a second sub-model, the MLP, to estimate the propagation. These two sub-models are inseparable as they are trained together.

In this approach, preparing the training data requires matching the RSS with the geographical location map, which is presented as 2D images. The map is divided into grids; for instance, in [34], the propagation area was divided into blocks of 10 × 10 square meters, where each block corresponds to a single pixel on the input image, and each pixel has a measured RSS value. In the majority of these approaches, the models are then trained using data that is simulated using various ray tracing software rather than real measurements. For instance, Sotiroudis et al [32] used ray tracing to create 35,395 pairs of images and RSS predictions, Similar strategy was also used to generate 56,000 radio simulation maps in [31,33], Xia et al. [4] used a combination of 800,000 ray tracing simulations and 400,000 empirical field data, and Masood et al. [34] used ray tracing, however, do not provide any information about the number of simulation samples.

The DL-based models were able to mimic the performance of the ray tracing simulations that served as the ground truth; however, the important achievement is the run-time of using DL-based models, which is two to three orders of magnitude faster [30].

DeepLoRa [35] uses a similar approach with a grid size of 10 × 10 square meters. However, to solve the path-loss regression problem, a bidirectional-long short-term memory (bi-LSTM) is used instead of an MLP. The authors conducted a measurement campaign and collected a total of 30,000 RSS indicators (RSSI); however, after preprocessing and griding, it was reduced to 4000 records. 

### 2.3. Comparison between DL and ML Approaches 

Reviewed studies indicate that AI-based models have delivered the prospect of increased accuracy compared to empirical models. DL-based models reduced run-time compared to ray tracing at the cost of time-consuming training phase [22]. AI-based models added adaptivity, where other impacting factors, such as LoRa’s SF and topographical information from the environment, could be integrated. However, both methods (ML and DL) have their disadvantages. There are multiple limitations to the current DL-based models.

#### 2.3.1. Training and Overfitting

The first issue is overfitting, which is intrinsic to models with a large number of adjustable parameters. This is further exacerbated when compared to the volume of training data, which is often negligible compared to the model parameters. The U-Net-based models typically have millions of parameters; this was up to 25 million parameters in [31]. Although authors in [33] created a custom model using a combination of CNNs and MLPs, which reduced the number of parameters to 4241, they only had 2392 samples for training. The ML-based models have fewer parameters and are less prone to overfitting. For instance, Sotiroudis et al. [32] has 300 parameters, and [26] has only about 30 parameters. In contrast to DL-based models, ML-based models decouple the process of topographical feature extraction from the regression task of propagation estimation. This decoupling simplifies data requirements and training processes. In fact, authors of DeepLoRa [35] have adopted a similar strategy to create a LoRaWAN-specific model, as they recognized that the cost of training a single DL-based model is higher.

#### 2.3.2. Dataset Origin and Integrity

The second limitation of DL-based models is data source mismatch, where data is rather simulated instead of being empirically collected. Tuning tens of millions of parameters requires enormous datasets, which has consequently led the researchers to resort to simulated data, as a large-scale measurement campaign would be time-consuming and expensive, defeating the concept of creating a predictive model. For instance, it took the authors of DeepLoRa about 4 months to collect 30,000 data samples [35]. Lowering the resolution of the images to reduce the model parameters and, in turn, the data volume should not be considered as a solution, as it has been reported to have an adverse effect on the prediction performance. Sotiroudis et al. [32] documented an increase in mean absolute error (MAE) as the image resolution was reduced. In contrast, ML-based models have fewer parameters and do not require enormous data for training, which can be collected empirically.

Consequently, DL-based models could be misrepresenting the propagation. As these models are trained by ray tracing simulation data, at best, they are mimicking the utilized algorithm, assuming they achieve perfect accuracy. They may have discrepancies with real-world propagation, do not capture the characteristics of novel technologies such as LoRa’s CSS modulation, and vary between different ray tracing simulations. Furthermore, the grinding process reduces the granularity and the accuracy of the prediction, as only one RSSI value can be associated with a single pixel. For instance, Liu et al. [35] collected 30,000 data points; however, as part of the griding process for landcover prediction, authors had to take an average of all the measurements within an area of 100 square meters, which reduced the samples to 4000, effectively not utilizing about 85% of the data for training. The use of simulation and griding degrades the originality of prediction. The reports do not provide further statistical analysis, such as examining the mean and standard deviation of the data within each grid, to justify whether averaging was the most accurate method to represent data in areas of the grid.

#### 2.3.3. Reliability and Accuracy

The accuracy and reusability of the DL-based trained models is another concern. Since DL-based models use simulated data, their accuracy depends on the performance of ray tracing simulations, resolution, and availability of geographical images, which could vary from 25 to 400 square meters. Geographical images of the environment may not be available, such as in [32], where authors created their own custom images. Additionally, low-resolution images and the griding process averaging the data could contribute to more uncertainty and inaccuracies in the DL-based models.

#### 2.3.4. Feature Engineering

Feature extraction is the main difference between the ML-based and DL-based approaches. While in DL-based models, feature extraction is off-loaded to DL algorithms (CNNs, U-Net), in ML-based models, this is performed prior to addressing the regression problem. The main contribution of DL-based models is the “black-box” approach to feature extraction, whereas machine learning techniques extract these features by means of deterministic algorithms such as image processing. ML-based models rely on pre-generated topographical features, which are not standardized. Currently, researchers extract these features using proprietary image processing algorithms applied to Google Maps, OpenStreetMap, or satellite images. Additionally, there is not an established set of extracted features, resulting in differences across the literature. This is partly due to a lack of quantitative understanding of the relevance and impact of these features, which must be addressed with further empirical studies in the future.

## 3. Implementation and Results

To develop an accurate and adaptive LoRaWAN model, a dataset is created using empirical measurements and topographical information to compare and test the suitability of ML-based algorithms.

### 3.1. Dataset and Topographical Features

This dataset contains 5007 RSSI entries collected over 13 square km of Glasgow City urban areas. Data is collected from three different base stations (BS1, BS2, BS3) communicating to a handheld mobile station (MS) that is equipped with a LoRa SX1272 module [36] and a global positioning system (GPS). In addition, topographical features are extracted at each measurement location using image processing techniques [10] and paired with the corresponding RSSI value. Details of these features are as follows:SF: LoRa’s spreading factor that varies from 7 to 12;Height difference: altitude difference between the transmitter and receiver;Lat, Lon, Alt: latitude, longitude, and altitude of the MS, acquired from GPS;FSPL: free-space path loss (FSPL) calculated $L_{0}=20\;log(\lambda /4\pi d)$ using the distance d between the BS and the MS, transmission wavelength λ=346.55 m, frequency 865.20 MHz; Clutter: calculated from the number of buildings on the Rx-Tx direct path;Road width: relative width of the street;AoA: acute angle of the LoS and the road axis;LoS ratio: clear the LoS ratio relative to d; this is a non-zero value when part of LoS is over the river;BS ID: base station identification BS-ID (BS1, BS2, BS3);Area Type: terrain type, (dense-urban = 1, urban = 2, open-urban = 3)

After encoding the BS ID, using one-hot-encoding, there are a total of 13 features in this dataset. Some of the parameters, such as the distance between MS and BS and radio frequency, are not explicitly present, and instead, they are calculated as the FSPL. Figure 1 provides a detailed representation of the BSs location, illustrating their respective distances, the designated measurement area, and the various area types. Area types 1 and 2 are distinctly color-coded. Any region within the measurement area that is not color-coded should be interpreted as area type 3, offering a clear visual distinction between the different zones.

### 3.2. Model Selection

This data analysis focuses on addressing the regression problem using empirically collected data and does not pertain to the use of DL for topographical feature extraction. This is for two reasons. The first is that data analysis utilizes empirically collected data rather than ray tracing simulation. The second reason is that topographical features are pre-extracted using image processing techniques rather than DL due to the size of the dataset. It is evident from the review of the literature that DL-based models require large volumes of data for training. Several different AI algorithms are chosen to cover the entire spectrum of AI, and different algorithms that have appeared in the literature for regression modeling of LoRa’s propagation are compared.

Optimized FSPL and linear regression (LR) are included to establish a baseline for capturing linear relationships in the data. However, LR is further expanded by using polynomial regression (PR), which is particularly useful as it can reveal the non-linear relationships in the dataset and examine the importance of feature engineering. Care must be taken to avoid overfitting, especially with high-degree polynomials, as the model might become too complex and sensitive to noise in the data.

In the ANN category, LSTM, MLP, and ANFIS were noted in the literature. MLPs are by far the most common model in use, either as a standalone model or as the last layer of DL-based algorithms. Simple architecture with three hidden layers can approximate any arbitrary continuous function, with the only downside being prone to overfitting, which can be avoided by the use of regularization and validating the training process. LSTM is beneficial for sequential data when dependencies between data points that are apart in time are of importance, such as time series, and they are irrelevant given this dataset. Adaptive neuro-fuzzy inference systems (ANFIS) can take advantage of an expert’s knowledge prior to the training and, therefore, can converge with fewer data samples; they can be helpful when conducting field measurements is costly. Two other classes are SVM/RVM and KNN; these models have the advantage of being less prone to overfitting and, therefore, easier to train compared to ANNs. The use of RVM was also noted due to its better generalization, where it was identified to suit the analysis of smaller-sized datasets. Finally, decision trees are chosen for being effective and having excellent AI explainability (XAI) [37]. Decision trees (DT) are also used as base estimators (weak learners) in the category of ensemble learning algorithms, such as random forest (RF) and gradient boosting (GR). GR and RF differ based on their training and aggregating the base estimators. Generally, the diversity of weak learners in ensemble learning algorithms provides less susceptibility to overfitting and better generalization.

### 3.3. Data Analysis and Result

Dataset is divided into training and holdout sets, each with 4007 and 1000 samples respectively. The training set is used for 5-fold cross-validation (CV) and hyper-parameter tuning. The holdout set is used for benchmarking the tuned models. All the algorithms are trained and tuned using a 5-fold CV, which involves multiple exhaustive hyperparameter tuning using grid search. Where possible, early stopping along with regularization were used to avoid overfitting.

To enable a comparative analysis of various algorithms, FSPL is used to establish a baseline against which the effectiveness of the other propagation variables that are extracted and used for machine learning can be compared. A regression analysis involving the pre-calculated FSPL is formulated in Equation (1), where $R\hat{SS}I$ is the model estimations.
(1)$$R\hat{SS}I=intercept+coefficient\times FSPL$$

In this optimized model, the FSPL’s parameters are intercept of 39.43 and a slope coefficient of −1.66, representing the best-fitted line. Figure 2 demonstrates this optimization, where the model achieved an RMSE of 9.01 dBm. Given the frequency of 865.20 MHz, the FSPL was calculated using the distance (d), which varied from 67.68 to 2275.36 m. The FPSL correspondingly spanned a range of 67.74 to 98.27 dB, depicted on the x-axis of Figure 2.

Next, linear Regression (LR) was applied to the dataset. Characterized by its simplicity and interpretability, linear regression aims to find the best linear fit by approximating a hyperplane to the data, presuming a linear relationship between all 13 of the propagation variables and the RSSI. This effectively forms a new propagation model by taking advantage of all the propagation variables. LR creates this new propagation model by linearly combining all the variables, and it comprises 14 coefficients. This is utilized as a rudimentary approach for estimating LoRa’s propagation and resulted in a root mean squared error (RMSE) of 8 dBm. This also demonstrates that added features, in addition to FSPL, can immediately improve the model’s performance.

Subsequently, polynomial regression (PR) was adopted to examine more complex relationships within the propagation variables. This facilitated the transformation of features into a higher-dimensional space, thereby allowing the model to form inherently nonlinear associations between these variables. Although polynomial regression is suited to addressing non-linear patterns, it incurs increased computational complexity due to expanding the number of features. Hyperparameter tuning resulted in a polynomial of degree 3, which transformed the 13 propagation variables into 560 features. A total number of features can be derived using the binomial coefficient in Equation (2), where *n* = 13 is the number of features, and *d* = 3 is the degree of the polynomial.
(2)$$\left(\begin{array}{c} n+d\;\\ d \end{array}\right)=\frac{\left(n+d\right)!}{n!\times d!\;}=\frac{16!}{13!\times 3!}=560$$

This expanded feature space suited the complexities inherent in LoRa’s propagation characteristics, as the RMSE was reduced to 6.65 dBm, while model parameters grew from 14 to 561.

Hyperparameter tuning examined many different MLP architectures, where the best performance was achieved from four hidden layers, each with seven neurons. The total number of parameters was, therefore, 274, making the model very lightweight, fast, and accurate. It was noted that the hyperbolic tangent activation function was selected by the tuning algorithm, which resulted in an RMSE of 6.85 dBm. ANFIS, with a 5-fold CV, achieved an RMSE of 7.16; however, when trained with only 30% of the data, it still maintained an accuracy of 8 dBm, making it a good choice for very small datasets.

The SVM achieved results similar to the MLP, with an RMSE of 7.03 dBm. The radial basis kernel function was used to map the data into a higher-dimensional space and handle the non-linearity of the data. RVM shares several similarities with SVM. Both models are known for their sparsity, meaning they rely on only a subset of the training data to make predictions. This sparsity is achieved through support vectors in SVM and relevance vectors in RVM. Additionally, both models employ the kernel trick to transform input features into a higher-dimensional space, enabling them to effectively handle complex, non-linear relationships in the data. RVM achieved an RMSE of 7.19 dBm and, by far, took the longest to train and tune. The number of model parameters in SVR and SVM are not pre-determined and depends on the number of support/relevance vectors. On average, trained SVM and RVM both had about 2000 parameters, mainly influenced by the number of support and relevance vectors identified during the training.

The application of the KNN yielded an RMSE of 6.7 dBm. Through the process of hyperparameter tuning, the optimal number of neighbors was determined to be 7. Additionally, a significant enhancement in the prediction accuracy was achieved by weighing the influence of each neighbor. This weighing was based on the inverse of their distance to the point being estimated. In this case, closer neighbors of a query point will have a greater influence than neighbors who are further away. KNN essentially stores or memorizes the entire dataset, and although providing good accuracy, it is computationally memory intensive, especially when the dataset has a considerable size. However, this is not a concern as the entire dataset (even including the holdout) is taking under 1.6 MB.

DT does not have model parameters like that of MLP, and the learning is embedded within the structure of the tree. This characteristic stands as one of the key advantages of DTs, as it renders the model more easily interpretable. The inherent transparency of DTs enhances their capabilities in XAI, offering deeper insights and a clearer understanding of the decision-making process. The tuned model achieved an RMSE of 6.54 dBm and had 188 leaf nodes and 187 branches. This DT is too large to visualize here; therefore, Figure 3 visualizes a pruned version of the original tree. Please note that since the tree is heavily pruned, some of the features are not present in this visualization. Furthermore, each node is color coded based on the value of the RSSI.

RF is an ensemble of multiple DTs, with each tree being trained on a bootstrapped random subset of the training set, and an averaging process is used to aggregate the output of all these trees for more accurate and stable results. RF is a good starting point for ensemble methods due to their simplicity and effectiveness. Tuned RF converged to an RMSE of 5.70 dBm with 100 estimators (decision trees). This was achieved without limiting the maximum depth of the DTs. GR was also tried, as it used DTs as its base estimator. However, GR creates DTs one at a time, where each new DT helps to correct errors made by the previous ones. The GR, with 79 estimators and a maximum depth of 7, achieved the best RSME of 5.53 dBm. Tree-based models, such as RF and GB, are known for their interpretability. This stems from their ability to naturally partition the feature space, which enables an in-depth examination of the impact and importance of each feature. Additionally, these models facilitate the analysis of partial dependencies, allowing us to understand how changes in feature values influence the target variable. This level of interpretability is particularly useful in gaining insights into how different features drive the model’s predictions. The importance and impact of the topographical features are demonstrated in the LoRa’s estimated propagation model in Figure 4. Where the location (latitude, longitude, and altitude) has the highest impact, followed by the height difference between BSs and the MS. SF has a 3% contribution to the estimation. The LoS ratio has the lowest contribution due to its sparsity; only 348 data samples had non-zero LoS ratio values.

Finally, partial dependence between the features and the RSSI is examined to identify how these features impact LoRa’s propagation and validate the feature engineering process. The partial dependence graph between the features and the RSSI is depicted in Figure 5. 

This graphical representation is useful for understanding how changes in a specific feature impact the RSSI in a quantitative manner. Graphs are generated from the tuned GR model as it achieves the best RMSE and quantifies the average dependence between the single feature and the RSSI. Therefore, the vertical axis shows the relative range of RSSI change, and the horizontal axis marks the deciles and shows the dynamic range of each feature. 

For instance, SF and the RSSI seem to have a linear but inverse dependence; SF 12 has a 1.5 dBm lower RSSI compared to SF 7. Suggesting that SF 12 can extend the reception of packets with 1.5 dBm lower RSSI. There is a direct dependence between the road width and the RSSI, where, on average, RSSI has increased by 1.5 dBm when transitioning from narrower to wider roads. The impact of the clutter is highly non-linear, and although initially attenuating the RSSI rapidly, it becomes ineffective after 20 buildings. When AoA (with respect to the road axis) is within 0 to 10 degrees, the RSSI is about 1.5 to 3 dBm higher, suggesting that the road could be acting as a waveguide.

## 4. Discussion and Conclusions

To summarize the benchmark, the Nash-Sutcliffe model coefficient (NSC) [38] and the RMSE values for all the models are compiled in Table 1. This comparison aims to assess the effectiveness of the various ML algorithms in developing a model for LoRa’s signal propagation. The NSC is utilized primarily because it is not sensitive to the dynamic range of the dependent variable, in this case, the RSSI. This attribute of NSC is particularly advantageous, as the sensitivity of one wireless technology can vary, therefore providing a more consistent and technology-agnostic measure. NSC ranges from 1 to −∞ , with NSC=1 indicating a perfect prediction, and NSC≤0 indicates that the model’s predictive power is no better than a simple mean of the RSSI. Finally, RMSE is provided as the most commonly reported measure in the literature. 

Focus of this article is not on asserting a general superiority of one machine algorithm over others; instead, it emphasizes the importance of careful model selection based on factors such as dataset size, tunable parameters, computational power, and specific problem requirements, including explainability of decision-making.

The article provided an overview of the cutting-edge developments in AI-driven LoRa propagation modeling. Different methods of extracting topographical features and embedding them in regression modeling algorithms were discussed and compared. Decoupling these two tasks via separate models alleviates the reliance on large simulation data, as even limited field measurement data proves to be adequate for effectively addressing the regression problem. Moreover, the utilization of deterministic terrain features in conjunction with tree-based models, particularly GR, marks a stride towards the realm of XAI. 

GR emerged as a superior choice in terms of accuracy, efficiency in tuning, training, and prediction speed. SVM was highly sensitive to the kernel coefficient optimization, and ANNs were highly sensitive to the number of hidden layers. However, GR achieved much better results out of the box with minimal effort. Its interpretability was instrumental in both quantitative and qualitative analysis of the propagation variables and their impact. Decoupling the regression and feature extraction and use of GR facilitated a deeper comprehension of the underlying mechanisms governing LoRa’s propagation. This unveiled a more nuanced understanding of these complex interactions, which is not possible through a black box feature extraction approach and is yet to be addressed with the further advanced techniques of the XAI.

Employing parametric (LR, PR) or tree-based (DT, RF, and GR) models with pre-extracted features directly enhances the explainability of the propagation model. Their inherent transparency and simplicity make them ideal for generating insights into complex model behaviors. In fact, these models are exploited by advanced techniques such as local interpretable model-agnostic explanations (LIME) [39] to provide AI interpretability for more complex models. LIME, in particular, leverages these models to create local, understandable approximations of predictions made by more complex algorithms [40]. This is essential in domains where understanding the ‘why’ behind a prediction is as crucial as the prediction itself.

## Figures and Tables

**Figure 1 sensors-24-00860-f001:**
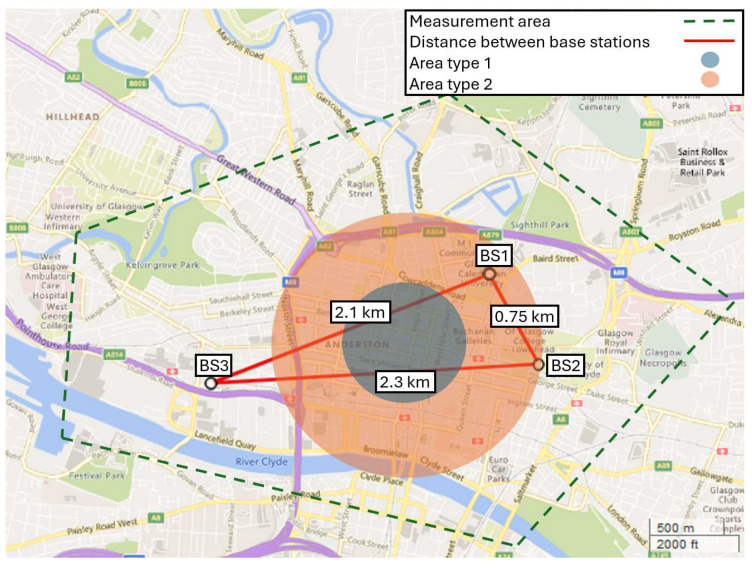
Area of experiment with location of base stations and area types.

**Figure 2 sensors-24-00860-f002:**
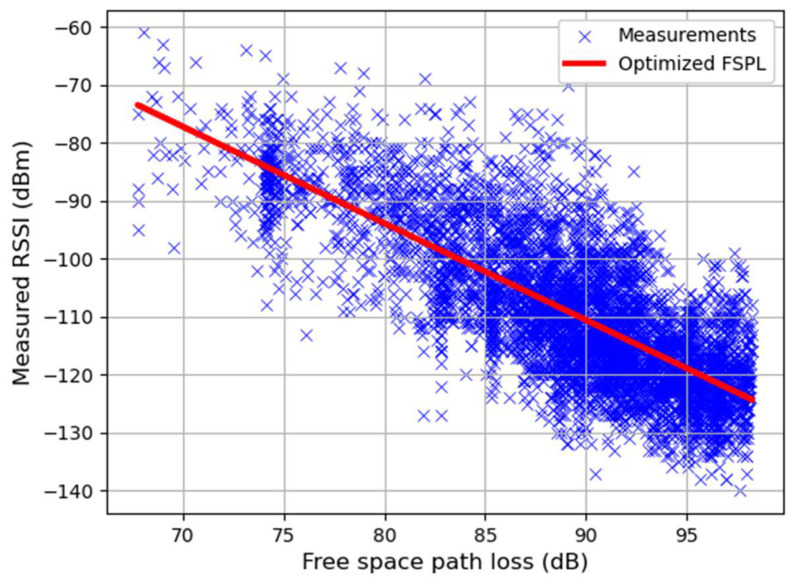
Comparing the optimized FSPL against measured RSSI.

**Figure 3 sensors-24-00860-f003:**
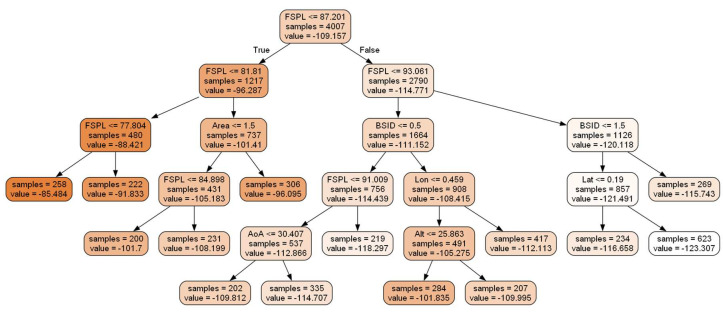
Visualizing a heavily pruned version of the trained DT model to estimate LoRa’s propagation.

**Figure 4 sensors-24-00860-f004:**
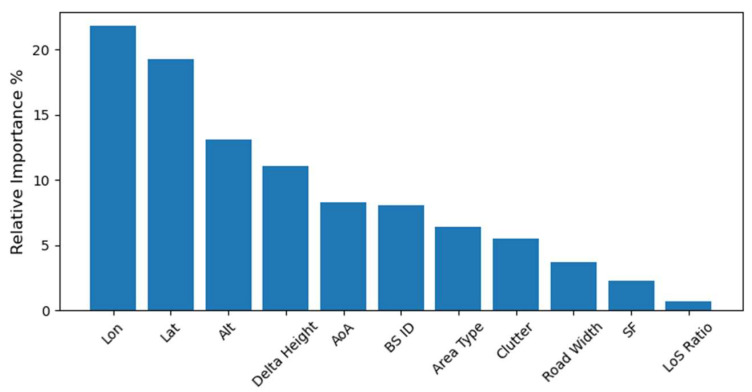
Impact and importance of environmental parameters to LoRa’s propagation estimation and modeling.

**Figure 5 sensors-24-00860-f005:**
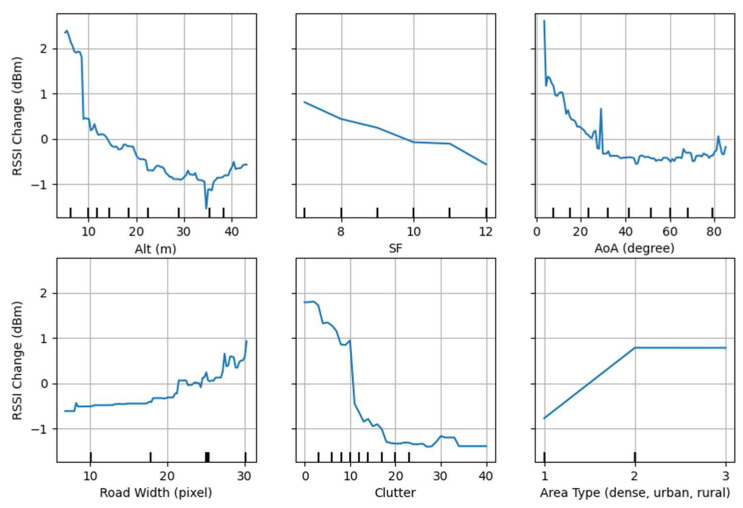
Partial dependence of RSSI on model features.

**Table 1 sensors-24-00860-t001:** Comparing RMSE and NSC tuned models using the holdout set.

Model		RMSE	NSC
Parametric regression	Optimized FSPL	9.01	0.25
	LR	8.01	0.37
	PR	6.65	0.67
Advanced nonlinear	MLP	6.85	0.62
	ANFIS	7.16	0.57
Kernel based	SVM	7.03	0.58
	RVM	7.19	0.57
Instance based	KNN	6.7	0.63
Tree based	DT	6.54	0.71
Ensemble of trees	RF	5.7	0.77
	GR	5.53	0.78

## Data Availability

Data is contained within the article.

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
