# Peer review of "Explainable Machine Learning for LoRaWAN Link Budget Analysis and Modeling"

_sensors, 2024, doi:10.3390/s24030860_

Round 1
Reviewer 1 Report
Comments and Suggestions for Authors
While the manuscript “Explainable Machine Learning for LoRaWAN Link Budget Analysis and Modeling” presents an intriguing exploration of the convergence of artificial intelligence and LoRaWAN network planning, the following technical rejection comments are provided for your consideration:
· The manuscript lacks a clear structure, making it challenging for readers to follow the flow of the research. Please consider revising the organization and improving the overall clarity of the document.
· A more detailed explanation of the data collection process, including sample size, data sources, and validation methods, is necessary for establishing the credibility and reliability of the empirical data used in the study.
· While the article mentions the use of decision-tree-based gradient boosting and reports the root-mean-squared error, a comprehensive discussion on the choice of evaluation metrics and a comparison with other relevant metrics would enhance the robustness of the analysis.
· The comparative analysis between different machine learning algorithms needs to be more comprehensive. Include a discussion on the strengths and limitations of each algorithm considered, providing a more balanced view for readers.
· While the article emphasizes the interpretability of the gradient boosting model, a deeper discussion on the interpretability of other models and a comparison of their transparency would add depth to the analysis, aligning with the focus on Explainable AI (XAI).
· The manuscript briefly mentions sensitivity issues with SVM and ANNs but lacks a thorough exploration of potential solutions or alternative approaches to mitigate these challenges. A more in-depth analysis of model sensitivity is required.
· Consider discussing the generalizability of the proposed model across different geographical locations, frequencies, or LoRaWAN deployment scenarios. This will enhance the practical applicability of the research.
· The manuscript could benefit from a more explicit discussion of the limitations of the proposed methodology, addressing potential constraints, assumptions made, and areas where improvements can be explored in future research.
· While the article provides a detailed account of the proposed methodology, it lacks sufficient integration with existing literature.
Author Response
Please see the attached word document

Reviewer 2 Report
Comments and Suggestions for Authors
The paper "Explainable Machine Learning for LoRaWAN Link Budget Analysis and Modeling" explores the use of artificial intelligence in planning LoRa networks. It investigates machine learning algorithms and empirical data to develop a propagation model for LoRaWAN. The authors propose separating feature extraction and regression analysis, which simplifies training data requirements. The decision-tree-based gradient boosting method they used showed the lowest error and interpretability benefits. The model offers insights into signal strength dependence on variables and improved sensitivity analysis.
To address issues in the paper, the authors might consider:
More diverse data could enhance the model's generalizability and robustness.
Comparing different algorithms could yield insights into the most effective methods for this application.
Validating the model's predictions in various real-world scenarios would strengthen its practical applicability.
A thorough investigation of errors made by the model might reveal specific areas for improvement.
Developing methods to better understand the model's decision-making process could be beneficial for practical applications.
Author Response
Please see the attached word document

Reviewer 3 Report
Comments and Suggestions for Authors
The paper is of interest to the journal and contains new scientific results.
Remarks:
1.What is the dimensionality of the ordinate axis in Figure 1? Probably dBm. It should be specified.
2.How were the measured values in Figure 1 obtained with RSSI less than -110 dBm? That is, after all, a complete absence of signal.
3.On page 8 lines 352-353 the authors write: "It was noted that the SVM was highly sensitive to the kernel coefficient, which made hyper parameter tuning a crucial step". Are there any quantitative estimates of this sensitivity?
4.On page 8 lines 358-359 the authors write: "RVM achieved an RMSE of 7.19 and by far took the longest to train and tune". Does this mean 7.19 dBm?
5.Figure 4 shows Partial dependence of RSSI on model features. RSSI Change values are plotted on the ordinate axis. From what absolute RSSI value are they plotted? Does this absolute value of RSSI somehow affect the appearance of the curves presented in Figure 4?
The work can be accepted after a minor revision.
Author Response
Please see the attached word document

Round 2
Reviewer 1 Report
Comments and Suggestions for Authors
It can be accepted.